# Development of a Control Strategy in an Isokinetic Device for Physical Rehabilitation

**DOI:** 10.3390/s23135827

**Published:** 2023-06-22

**Authors:** Jorge Andrés Peñaloza-González, Sergey González-Mejía, José Isidro García-Melo

**Affiliations:** School of Mechanical Engineering, Faculty of Engineering, Universidad del Valle, Cali 760042, Colombia; sergey.gonzalez@correounivalle.edu.co (S.G.-M.); jose.i.garcia@correounivalle.edu.co (J.I.G.-M.)

**Keywords:** dynamic modeling, electric brake control, isokinetic, isokinetic control, dynamometry, rehabilitation

## Abstract

Robotic-assisted rehabilitation is currently being applied to improve the effectiveness of human gait rehabilitation and recover the mobility and strength after a stroke or spinal cord injury; a robotic assistant can allow the active participation of the patient and the supervision of the collected data and decrease the labor required from therapists during the patient’s training exercises. The goal of gait rehabilitation with robotic-based assistance is to restore motor function by using diverse control strategies, taking account of the physical interaction with the lower limbs of the patient. Over the last few years, researchers have extracted useful information from the patient’s biological signals that can effectively reflect movement intention and muscle activation. One way to evaluate progress in rehabilitation is through isokinetic prototype tests that describe the dynamic characteristics of an isokinetic leg extension device for rehabilitation and control action. These tests use an isokinetic system to assess muscle strength and performance in a patient during isometric or isokinetic contraction. An experimental prototype shown in the following work allows the device’s performance to be evaluated in a controlled environment before the patient’s use. New features provide a control system that can be teleoperated for distributed structures, enabling the remote operation and management of the device. In order to achieve physical recovery from musculoskeletal injuries in the lower limbs and the reintegration of the affected subject into society as an independent and autonomous individual in their daily activities, a control model that introduces a medical isokinetic rehabilitation protocol is presented, in which the element that carries out such protocol consists of a magnetic particle brake whose control action is strongly influenced by the dynamics of the system when in contact with the end user—specifically, the patient’s legs in the stretch from the knee to the ankle. The results of these tests are valuable for health professionals seeking to measure their patient’s progress during the rehabilitation process and determine when it is safe and appropriate to advance in their treatment.

## 1. Introduction

An isokinetic testing protocol based on an experimental prototype is an approach to evaluate the effectiveness of a device or treatment by using a prototype and an isokinetic testing system. The isokinetic concept refers to the ability of the system to maintain a constant speed during muscle contraction, allowing for accurate measurement of strength and performance [1]. Moreover, an experimental prototype allows the device’s performance to be evaluated in a controlled environment before patient use.

Isokinetic rehabilitation is a technique used to recover muscle injuries and other conditions that affect the ability to move and perform physical activities. This technique uses isokinetic devices, allowing muscles to be exercised at a constant speed and measured torque. It has been shown to be effective in recovering injuries, allowing patients to exercise safely with controlled intensity, facilitating muscle strength and endurance recovery. This technique can also help prevent muscle wasting and atrophy in patients with physical disabilities or prolonged periods of inactivity [2]. Other research recommends the integration of these technological advances in post-surgical rehabilitation processes or in the development of protocols in sports medicine [3].

This research aims to design a control system for an isokinetic rehabilitation protocol to recover leg muscle injuries. For this purpose, a study of the dynamic components involved in lower limb extension will be performed using the rehabilitation device from Universidad del Valle, seen in Figure 1, which implements a magnetic particle brake to restrict the extension movement; it has speed control and serves as a model to facilitate the demonstration of the effectiveness of isokinetic rehabilitation and the recovery of muscle strength and endurance in patients with leg muscle injuries; the device is similar to an isokinetic dynamometer with the difference that it has a robustness that facilitates a customized configuration [4].

The leg rehabilitation mechanism seen in Figure 1 consists of several key components. The main structure is made of high-strength materials, utilizing aluminum and stainless steel. These materials provide durability and lightness, essential for facilitating movement and device portability. Moreover, bronze bushings and shafts ensure smooth and friction-free operation, optimizing the mechanism’s efficiency [5].

Regarding the electrical power supply, the system is connected to a standard 110-volt (V) power source, which is then reduced to 24 V to power the mechanism’s power systems. An additional conversion is used to obtain a 5 V voltage for device control. This configuration ensures a stable and safe electrical supply, guaranteeing optimal performance and protecting the electronic components. The control interface of the mechanism has been developed using the PHP programming language. This interface integrates with an intuitive touchscreen, enabling simple and precise device control. Users can adjust specific parameters, monitor exercise progress, and make custom configurations based on individual needs. Furthermore, thanks to its portable and adaptable design, the mechanism is compatible with teleoperation medical technologies, allowing for remote monitoring and efficient patient supervision in real-time [6].

Physical therapists often use Isokinetic exercise as a relatively safe and effective way to strengthen certain muscles in a person recovering from an injury. Some of its benefits include experiencing resistance throughout a joint’s entire range of motion, controlled resistance, and speed to reduce the risk of injury; and others, such as physical therapists, can measure a person’s progress accurately. According to a prior study, isokinetic exercises have been shown to be more effective in improving functional performance than isometric and isotonic exercises [7].

Additionally, this type of exercise may be a better form of muscle recovery than other types. A study [8] showed that it was more effective in athletes with lower back pain than standard rehabilitation exercises. Physical therapists can modify isokinetic exercises to suit people of different abilities, making this training more accessible and appealing. Other studies [9] concluded that isokinetic workouts were the best at increasing isokinetic strength, an important variable to supervise under a rehabilitation process by healthcare personnel. In this sense, it is advisable to integrate these technological advances with information technologies to improve the implementation of telemedicine programs in physical rehabilitation [10,11].

An isokinetic dynamometer is used in sports, exercise, and clinical settings to measure muscle strength, torque, and power. The device allows specific testing on different joints and muscle groups, adapting to different speeds and ranges of motion. Isolation of the joint of interest allows a complete evaluation to compare right and left limbs, detecting potential injury risks or areas for improvement.

Isokinetic dynamometry is used to evaluate and strengthen limbs by measuring maximum force at a specific speed during movement; dynamometry systems allow comparison of concentric and eccentric muscle contractions while maintaining a constant speed; they are commonly used in physical therapy clinics for patients who have suffered injuries or are in the post-surgical state, making comparisons between the affected and non-affected limb. The percentage of the strength of the non-affected limb is used to determine when a patient can return to physical activity. In countries such as the United States, these systems are still used to assess limb strength, although they were used more frequently in the past [12].

Regarding assessing muscles across a full range of movements, isokinetic testing has several advantages over isometric testing. Isokinetic testing allows the health specialist to analyze agonist–antagonist relationships, determine maximum strength, and evaluate endurance and other indicators of muscle capacity. Through specialized computer tools, it is possible to compare two limbs and provide feedback during the test, delivering valuable information on the degree of recovery of average strength. For patients with extreme cases of weakness, evaluations can be performed with isokinetic devices using torque values obtained through comparison between passive isokinetic motion and concentric maximum effort movement, two of the most significant variables in this type of test [13].

Dynamometry is a technique used to measure muscle strength, which includes different functional and manual muscle tests. Functional tests are easy to perform and do not require equipment, but they have limitations in their exact relationship to muscle strength. Manual muscle testing is a technique used in neurological examinations but requires an experienced examiner and has high variability due to evaluators. Isometric dynamometry measures static muscle strength, while isokinetic dynamometry measures dynamic muscle strength. These techniques provide accurate information about the muscle strength of all major muscle groups of the upper and lower extremities. However, comparisons between different studies are rarely made due to the lack of standardization of test procedures and different equipment in laboratories and neuromuscular centers [14].

Some studies have evaluated the reliability of different dynamometers in measuring maximum torque in knee flexion and extension in athletes; the results showed that knee maximum flexion and extension tests performed in specific contractions with different times of use had similar values; this indicates that the measurement of muscle strength of the dynamometer does not change over time. Therefore, the hypothesis is not supported; the study suggests that the IsoMed-2000 isokinetic dynamometer, owned by D. & R. Ferstl GmbH located in Hemau, Germany, is suitable for evaluating muscle strength and changes in athletes, despite some measurement errors; it can be used in physiotherapy and rehabilitation clinics and for athletes in injury risk analysis and muscle strength development. However, it is not reliable for measuring knee flexion torque. Furthermore, this study also highlights that the control system of the IsoMed-2000 dynamometer may not be robust enough. As a result, the dynamometer may not be reliable in certain conditions or for specific populations. It is essential to be mindful of this limitation when using the device to evaluate muscle strength and to be careful when interpreting the results. It would be beneficial to consider alternative methods or equipment with a more robust control system for measuring knee flexion torque [15]. Several rehabilitation solutions are available on the market, but few focus on developing standard isokinetic exercises. Most solutions focus on traditional physical therapy, such as physiotherapy, occupational therapy, and pain therapy. These solutions are based on strengthening and stretching exercises and may include equipment such as elastic bands, weights, and cardio machines. Although these solutions can effectively improve mobility and strength, they do not specifically focus on developing isokinetic exercises, and their control is limited to the device’s dimensions.

Isokinetic exercises are a specific rehabilitation approach based on using isokinetic machines to exercise muscles at a constant speed. This approach is considered more accurate and safer than traditional exercises, allowing for a more precise measurement of strength and performance [16]. However, although isokinetic exercises are a valuable technique for rehabilitation, they are not as common in the market because they require specialized equipment and trained personnel. Commercial isokinetic dynamometry systems are specialized devices that measure muscle strength and performance in patients with muscle injuries. However, devices such as the System 4 Pro™ [17], owned by Biodex located in Shirley, NY, or the Cybex 6000/HUMAC [18], owned by Computer Sports Medicine Inc., located in Stoughton, MA, are insufficient to apply a complete isokinetic rehabilitation protocol, as they do not include all the necessary components for a complete treatment. These types of systems require direct supervision from medical staff, which increases the cost of treatment and limits the number of patients that can be treated in one day; this makes treatment more expensive and less accessible for patients [19]. These devices are also insufficient to apply a complete isokinetic rehabilitation protocol. They focus mainly on strength evaluation and do not include a complete exercise and therapy program for injury recovery [20].

The study [21] compared two machines, the SMM, owned by iMoment, located in Maribor, Slovenia, and the Biodex System Pro™ 4, regarding their reliability in measuring knee flexion and extension at different speeds. The study concluded that the differences between the machines were insignificant and that most of the variations in results could be attributed to biological differences between subjects and the difficulty of repeating the same performance on different occasions [22]. However, some technical differences were identified that could impact the results, such as the damping system and seat fixation. The authors found that the machines modulate between high and moderate absolute reliability compared to other studies [23]. The study suggests that the manufacturer of the SMM iMoment machine improves the software to validate control objectively by adjusting the parameters to the patient’s dimensions, changing the mechanism and simplifying the movement and fixation system [24].

Additionally, these devices are expensive and not very durable, requiring a significant investment in equipment and qualified personnel. Moreover, they do not consider the dynamics of the mechanisms and the patient for isokinetic control, which limits their effectiveness in muscle injury rehabilitation. It is important to continue researching and developing solutions to improve the accessibility and effectiveness of isokinetic exercises in rehabilitation. In summary, although isokinetic exercises are a valuable technique for rehabilitation, current solutions on the market do not aim to develop this technique due to the lack of availability, cost, and access to the necessary equipment and trained personnel. However, it is important to continue researching and developing solutions to improve the accessibility and effectiveness of isokinetic exercises in rehabilitation.

In [25], a catalog that classifies exoskeletons by category (commercial, industrial, medical, and military) and by application (assistive, augmentation, pediatric, rehabilitation, research, etc.) is presented; this exoskeleton report is focused on supplying news and resources on the emerging technological field of exoskeletons, exosuits, and wearable robotics. Moreover, the systematic review [26] summarizes the main technical aspects of 25 wearable lower-limb exoskeletons, such as commercial availability, assisted joints, type of population, control method, power storage, gait phase detection, gait initiation mode, use (rehabilitation/augmentative), certification (CE mark/FDA approval), target user, etc. The paper [27] documents a review of robotic devices for upper limb rehabilitation, in which several issues are discussed, such as application field, target group, type of assistance, mechanical design, control strategy, and clinical evaluation; it includes a comprehensive, tabulated comparison of technical solutions implemented in various systems. Other similar works are presented in [28,29]. The work [30] describes the blood pressure and heart rate responses triggered by an isokinetic testing protocol in professional soccer players and compares cardiovascular parameters at the completion of the isokinetic protocol with those during a treadmill test.

This document delves into the development of an isokinetic rehabilitation control system, encompassing various aspects of its architecture and functionality. The subsequent sections explore different components and considerations involved in the system. Section 2, “Architecture,” provides an overview of the system’s overall design and structure. Section 2.1 further delves into specific physical variables pertinent to the rehabilitation process. Section 2.2 discusses considerations for calculating the angle Φ, highlighting its significance in the system’s operation. Section 2.3 focuses on the calculation of the patellar tendon force components. Section 2.4 delves into calculating the torque the tendon generates along the leg. The characterization of the magnetic brake is explored in Section 2.5, emphasizing its role in the system’s functionality. Additionally, Section 2.6 delves into the considerations for controller design. Section 3, “Results and Discussions,” presents the findings and analysis of the implemented rehabilitation system. Finally, Section 4, “Conclusions,” concludes the document by summarizing the key findings, implications, and potential future directions for the rehabilitation system.

## 2. Architecture

Characterizing a device for isokinetic rehabilitation of the leg with architecture (see Figure 1) requires careful consideration, such as the dynamic equations of motion, the angle between the patellar tendon and the tibia, the calculation of the force components of the patellar tendon, the torque generated by the tendon along the leg, the characterization of a magnetic brake, and the design of a controller. These factors are critical to ensureing effective and safe rehabilitation of patients. For example, the accuracy in the calculation of the patellar tendon force will enable a good estimation of the generated torque. At the same time, proper characterization of the magnetic brake will guarantee suitable control of the resistance during the exercise. Furthermore, the design of an efficient controller will be crucial to ensure an adequate and safe response to changes in leg position and speed during rehabilitation.

### 2.1. Definition of Physical Variables Involved in the Rehabilitation System

In a testing scenario, the kinematic and dynamic characteristics of the bodies interacting within the system must be considered to achieve rehabilitation through techniques that allow for controlling the speed of a patient’s leg extension.

During lower limb rehabilitation processes, particularly in the case of the knee and leg, activities of limb extension are performed in which the body posture remains seated with a slight inclination of the back. The goal is to extend the leg from a reference position of 0° when the leg is perpendicular to the ground and extend it to achieve an inclination of 90° while keeping the axis of the leg parallel to the ground [31]. Figure 2 shows the behavior of an uncontrolled leg extension. The angular position change is seen in Figure 2a, and the angular speed change of a leg during the extension initiated at 0 radians and extended to approximately 1.5 radians is seen in Figure 2b.

To improve this pre-process and the rehabilitation process, it has been suggested to implement a feature in which the patient performs the extension process with an equal speed throughout the entire arc of motion, as ensuring the speed requires the body to control the force with which the leg is lifted and, in turn, facilitates the collection of information on the patient’s activity. Figure 3 shows an ideal leg extension with constant speed through the rehabilitation exercise.

The considered variables involved in each subsystem are the following: $\tau_{P}$ is the torque due to (x,y) components of the force $F_{p}$ on the patellar tendon; $\tau_{WL}$ is the torque due to the weight of the leg; $\tau_{PB}$ is the torque due to the pedal of the mechanical brake system; $\tau_{B}$ is the torque opposing the movement generated by the magnetic brake; $\tau_{K}$ is the torque observed at the knee reference point concerning the system; $M_{L}$ is the mass of the leg; $\omega_{F}$ is the current angular speed at the time of measurement in rads; $\omega_{0}$ is the previous speed at the time of measurement in rads; t is the sampling time; $L_{L}$ is the length of the leg; θ is the angle measured in radians of the position of the leg upright relative to the ground and the thigh that is in a horizontal position to the ground with a reference point such as the knee, k; g is gravity acceleration equivalent to 9.81 ms2; $M_{PB}$ is the mass of the machine pedal; $d_{PB}$ is the distance of the pedal from the axis to the point of contact with the leg; α is the angular acceleration; I is the moment of inertia calculated as $I=mr^{2}$; Φ is the angle measured in radians generated between the patellar tendon and the tibia; V is the voltage applied to the brake system; R is the resistance of the brake system for an applied voltage and dependent on the system current for a given moment; and $K_{B}$ is the motor constant of magnetic dipole moment expressed in engineering units as NmA.

Within the dynamic of the patient–machine system, the sum of torques is equal to the angular movement acceleration, α, by the leg inertia, I; that is, ${\Sigma \tau }_{K}=I\alpha$. Equation (1) shows the sum of torques expressed in the individual torques for each component considered in the rehabilitation device system. The generated counterclockwise torques are the ones produced by the patellar tendon force, $\tau_{P}$, and the torque due to the movement of the leg when lifting, $\tau_{K}$, are represented positively. In contrast, in a clockwise direction, the torque due to the weight of the leg, $\tau_{WL}$, the weight of the pedal, $\tau_{PB}$, and the opposing torque induced by the brake, $\tau_{B}$, when actuated, are present.
(1)$${\Sigma \tau }_{K}={\tau_{WL}+\tau_{PB}+\tau_{B}-\tau }_{P}=I\alpha +\tau_{C}+\tau_{g}=\tau_{net}$$

The vector [$\tau_{C}$, $\tau_{g}$, $\tau_{net}$] represents the torque due to Coriolis and centrifugal forces, the torque due to gravity, and the net torque at the knee joint, respectively. Figure 4 shows the extended leg experiments with the torques produced by the gravitational forces due to the device on the pedal weight $T_{Mg}$ and the human leg weight $T_{Hg}$ and how they interact negatively on the opposite direction of the knee torque.

At the knee joint, a human knee with the leg at the maximum extension experiences torque reduction due to the gravity and the sum of the inertia moments of the angular acceleration and the inertia moments of the human leg and the device pedal, which are ignored due to the minimum impact to the exercise. Figure 4 reactions at the knee joint are shown in Equation (2):(2)$$\begin{array}{cc} \tau_{K} & =\left(I_{H}+I_{M}\right)\alpha +\tau_{Hg}+\tau_{Mg}\\ & \tau_{K}=\tau_{Hg}+\tau_{Mg}. \end{array}$$

For phases when the leg is in the movement of being extended, the knee joint is experiencing dynamic variables due to the anthropomorphic values, which can be seen in Figure 5.

In the maximum leg extension seen in Figure 4, the radial acceleration becomes zero because the leg stops moving upon extension. In the case presented by Figure 5a, the sum of torques due to gravity for the human, $\tau_{Hg}$, and machine $\tau_{Mg},$ are counterbalanced by the torque generated by the patellar tendon force $\tau_{P},$ going in the opposite direction; for this case, $\tau_{k}=\tau_{P}$.

A simplified model is proposed to derive the system torques based on the following patient measurements as seen in Figure 5b. In this case, the depth of the tendon within the leg is considered, forming a right-angled triangle that can be vectorially decomposed, as Figure 6 indicates.

To facilitate the following calculation, it is estimated that the value of the center of gravity of the leg is exactly half the length of the leg. Using the Equation (2), and knowing that τWL=12LLMLgsinθ, τPB=12MPBgdPBsinθ and τB=VRKB, the torque due to the moment of inertial in the knee joint is Equation (3):(3)$$\tau_{k}=I\alpha =M_{L}L_{L}^{2}\left(\frac{\omega_{F}-\omega_{0}}{t}\right)sin\left(\theta \right).$$

Next, the force $F_{p}$ in the patellar tendon is solved as Equation (4):(4)$$F_{P}=\frac{\frac{L_{L}M_{L}g}{2}sin\left(\theta \right)+\frac{M_{PB}gd_{PB}}{2}sin\left(\theta \right)+\frac{V}{R}K_{B}-M_{L}\left(\frac{\omega_{F}-\omega_{0}}{t}\right)L_{L}^{2}sin\left(\theta \right)}{L_{L}(Y-component\;of\;F_{P}+X-component\;of\;F_{P})}.$$

### 2.2. Considerations for Calculation of Angle Φ

Based on [32,33], for tendon length, $L_{T}$, the variations are approximately from 1/12 of the length of the leg, which is measured from the center of the knee to the foot’s sole when the leg is fully extended (θ=90°), to 1/11 of L when the leg is flexed (θ=0°), and from the surface of the knee locating the patellar bone to a horizontal reference to the fixation of the patellar tendon with the tibia bone with an approximate value of 1/55 of the length of the leg, $L_{PP}$. Figure 7a proposes a simple decomposition of the dimensions of the length of the patellar tendon when the leg is fully extended and fully flexed for the extension exercise, which generates a lineal behavior that permits us to find the relation between the patient’s leg length and the length of the patellar tendon. A description of the length relationship can be used to propose a point in which the extended and flexed leg length meets over the tibial bone, thus establishing the compression angle Φ as seen in Figure 7b.

For a leg of length $L_{L}$, the patellar tendon distance ($L_{T}$), as a function of the angle θ, is seen in Equation (5):(5)$$L_{T}=\frac{L_{L}}{11}\left(1-\frac{\theta }{6\pi }\right).$$

The angle Φ generated by the patellar tendon, and the adjacent side is calculated in Equation (6):(6)$$\Phi_{rad}=arcsen\left(\frac{1}{5\left(1-\frac{\theta_{rad}}{6\pi }\right)}\right).$$

Likewise, using the trigonometric identity $csc\left(arcsen\left(x\right)\right)=\frac{1}{x}$, Equation (6) can be viewed as $\theta_{rad}$ as a function of angle $\Phi_{rad}$ shown in Equation (7):(7)$$\theta_{rad}=6\pi \left(1+\frac{csccsc\left(\Phi_{rad}\right)}{5}\right).$$

### 2.3. Consideration for Calculation of the Patellar Tendon Force Components

The patellar tendon is responsible for exerting the force that generates the torque capable of lifting the lower leg during extension; the length of the patellar tendon is calculated based on the angle Φ and is expressed as LT=LL55cscΦrad. Considering a right triangle with the upper vertex at the lower contact with the patella and the furthest end at the rigid anchor point with the tibia (as seen in Figure 8), the distances to a point perpendicular under the patella and horizontal to the anchor point of the patellar tendon can be calculated, which facilitates the subsequent calculation of angle Φ.

Knowing that the side perpendicular to the angle Φ is LPP=LL55, the horizontal side ($L_{HP}$) to the angle Φ is calculated using the Pythagorean theorem as LHP=LL55Φrad. The components on the patellar tendon forces, seen in the Figure 9, are $F_{PY}=F_{P}cos{(\theta }_{rad}-\Phi_{rad})$ and $F_{PX}=F_{P}sin{(\theta }_{rad}-\Phi_{rad})$.

Therefore, the torque component, $\tau_{p}$, due to the patellar tendon force is calculated in Equation (8):(8)$$\tau_{P}=F_{P}\frac{L_{L}}{55}\left(\Phi_{rad}\right)\left(sin{(\theta }_{rad}-\Phi_{rad}\right)+cos{(\theta }_{rad}-\Phi_{rad})).$$

The component of the distance used to calculate the torque in Equation (8) can be extracted and denoted as $x_{P}=\frac{L_{L}}{55}\left(\Phi_{rad}\right)\left(sin{(\theta }_{rad}-\Phi_{rad}\right)+cos{(\theta }_{rad}-\Phi_{rad}))$. Then, the distant component of $F_{P}$ seen on Equation (5) is replaced by $x_{P}$, generating Equation (9) as follows:(9)$$F_{P}=\frac{\frac{L_{L}}{2}M_{L}gsin\left(\theta_{rad}\right)+\frac{d_{PB}}{2}M_{PB}gsin\left(\theta_{rad}\right)+\tau_{B}-M_{L}L_{L}^{2}\left(\frac{\omega_{F}-\omega_{0}}{t}\right)sin\left(\theta_{rad}\right)}{\frac{L_{L}}{55}csc\left(\Phi_{rad}\right)\left(sin\left(\theta_{rad}-\Phi_{rad}\right)+cos\left(\theta_{rad}-\Phi_{rad}\right)\right)}.$$

For a particular case, the supply voltage to the brake is zero and there will be no counter-torque; therefore, $\tau_{B}=0\;Nm$. Thus, replacing $\tau_{B}$ into Equation (9) gives us Equation (10):(10)$$F_{P}=\frac{\frac{L_{L}}{2}M_{L}gsin\left(\theta_{rad}\right)+\frac{d_{PB}}{2}M_{PB}gsin\left(\theta_{rad}\right)-M_{L}L_{L}^{2}\left(\frac{\omega_{F}-\omega_{0}}{t}\right)sin\left(\theta_{rad}\right)}{\frac{L_{L}}{55}csc\left(\Phi_{rad}\right)\left(sin\left(\theta_{rad}-\Phi_{rad}\right)+cos\left(\theta_{rad}-\Phi_{rad}\right)\right)}.$$

The difference of the current speed with respect to the previous one over the sample time can be replaced by angular acceleration, $\alpha_{rad}$, as follows in Equation (11):(11)$$F_{P}=\frac{\frac{L_{L}}{2}M_{L}gsin\left(\theta_{rad}\right)+\frac{d_{PB}}{2}M_{PB}gsin\left(\theta_{rad}\right)-M_{L}L_{L}^{2}\alpha_{rad}sin\left(\theta_{rad}\right)}{\frac{L_{L}}{55}csc\left(\Phi_{rad}\right)\left(sin\left(\theta_{rad}-\Phi_{rad}\right)+cos\left(\theta_{rad}-\Phi_{rad}\right)\right)}.$$

Then the torque applied by the patellar tendon is $\tau_{P}=F_{P}x_{P}$.

### 2.4. Considerations for Calculation of Torque Generated by the Tendon along the Leg

The torque generated by the tendon during leg extension can be obtained by simplifying the model of the leg seen in Figure 10, considering the tendon force as a beam suspended over a pulley symbolizing the knee and the origin point for measurement.

Considering the vertical directions of the force exerted by the patellar tendon in the upward direction and the force caused by the weight of the leg in a downward direction, the human torque, seen in Equation (12), can be expressed as the difference between these two components as follows, taking as reference the distances from the knee to the tendon anchor point and half the distance of the leg to calculate the weight due to gravity:(12)τH=τP−LL2WLsinθrad+MLLL2αradsinθrad,τH=FPLL55cscΦradsinθrad−Φrad+cosθrad−Φrad+⋯−LL2MLgsinθrad+MLLL2αradsinθrad.

### 2.5. Considerations for Magnetic Brake Characterization

For the case of the brake, when the system is working in the off condition, it does not generate an opposite torque to the leg movement; therefore, at the initial phase of the protocol, it is assumed that no torque is applied to the system. On the other hand, when the brake is in the on condition, the voltage applied to the electrical system that makes the brake generate an electric current is conditioned by the electrical structure of the selected brake [18], and for this case, it can be reduced to a constant that involves the brake material, circuits, and dimensions, which is denominated $K_{B}$. This constant with the interaction of the electrical conditions of voltage and resistance of the circuit produces a torque, τB=VRKB, and it is opposite to the torque generated by the extension of the leg, $\tau_{K}$. 

Regarding the leg moment of inertia seen in Equation (4), a patient’s leg placed in the rehabilitation device pushing on a weight sensor, which captures the mass ($M_{s})$, is measuring the force executed by the extension when the brake is in the on condition. The torque of the excited leg will be equivalent to the difference of the moment of inertia with the torque generated by the brake as seen in Equation (13)
(13)$$M_{s}L_{L}g=\frac{1}{3}M_{L}L_{L}^{2}\left(\frac{\omega_{F}-\omega_{0}}{t}\right)-\frac{VK_{B}}{R}.$$

Regarding the particle brake seen in Figure 1b, the Figure 11 shows the relationship between the generated torque and supplied current [18]; this can be used to infer a linear relationship between them.

From the behavior of the curve from Figure 11, the correspondent points are taken to be used as a basis for a linear regression algorithm to obtain an equation that describes the change in the total percentage of braking torque as a function of the applied current; therefore, $\%I=6.9408+0.9533\tau_{B}.$ Using the brake manufacturer’s data, it is possible to determine the torque generated by the magnetic brake as a function of the current consumed by the system at a given time, so the following expression is obtained: $\tau_{B}\left(V=24\;V\right)=0.8\;K_{B}=6\;Nm$. Therefore, the magnetic brake constant implemented in the system is KB=7.5 NmA. Due to the linear behavior of the current, in the case of the electrical voltage at 12 V, the brake torque is $\tau_{B}\left(V=12\;V\right)=3\;Nm$. The internal resistance of the element due to the physical and electrical components is R=30 Ω.

### 2.6. Controller Design Considerations

Elements are required that can measure the state of the system in a required instant; for the rehabilitation device, this requirement is solved by using a position sensor. This sensor is calibrated with an initial position for an extension movement when the leg is attached to the system and at rest, generating a right angle between the thigh and the leg in an upright position as seen in Figure 12. Position measurements can be used in conjunction with the sampling time to evaluate the speed of the extension movement.

The element to be controlled is the particle brake—specifically, its power supply system—and it is controlled performing an electrical resistance function; the amount of voltage that reaches the brake is manipulated and, in the same way, the action of the brake in the form of counter torque is modulated according to the speed parameters to be controlled. The controller objective is to modify the impedance for the voltage source and modulate the counter torque of the brake, $\tau_{B}$, to force the patient’s leg extension speed to keep its value under the threshold of the speed set by the device operator.

A proportional action for a rehabilitation device extension speed controller can be understood as the relation between the evaluated speed and the threshold speed and the voltage supply needed to feed the brake; so, the difference of the speeds remains positive, which means that the evaluated speed keeps its value under the threshold, in which case, the brake is in the off condition. The controller action increases or decreases the gain or the value of the brake resistivity according to the speed difference, also known as the error signal respect to time, $e\left(t\right)$. In case the speed difference evaluates a negative value, which means that the evaluated speed is over the threshold, the controller increases the gain of the voltage source to the brake circuit by increasing $\tau_{B}$. Consequently, a control action, $u\left(t\right)=K_{p}e\left(t\right)$, proportional to the angular speed error, e(t), is deployed, with $K_{p}$ being the proportional gain [35].

With this application, a quick response is achieved when the control action must generate rapid brake resistance; however, if the speed is close to the set point and the response is exceeded, it generates a delay in the response of the system. To improve the performance of the control action, there is an additional integrative control [36]. This type of control seeks to reduce and eliminate the steady-state error, which was present in the previous case when a slow response from the controller with low gain was required; in this case, the error signal is integrated during the period of the control action, so the response of the controller is delayed by the integrator constant, $K_{i}$, this directly affects the proportional delay and is represented in Equation (14):(14)$$u(t)=K_{i}\int_{0}^{t}e(t)dt+K_{p}e(t).$$

## 3. Results and Discussions

Figure 13 shows a dynamic system which involves the subsystems in the proposed rehabilitation exercise of a leg extension patient. The input block *pos_ref[rad]* is an array of the values taken during an uncontrolled human leg extension; this measure is used by the system as the signal that defines the angular position of the leg during the extension, Theta[rad]. The block *Fn_Phi_rad* is an operator that converts the Theta[Rad] signal into the Phi[rad] signifying the calculation of phi. The input block *vel_ref[deg/s]* is an array of the threshold value for the speed limit selected by the device operator; in the planted scenario, its value is set as reference speed of 60 degs; this signal is transformed to rads, used by the error calculation for the control subsystem. Function blocks are implemented to calculate the torque signals *T_WL[Nm]* for the torque generated by the leg weight; *T_WPB[Nm]* for the torque generated by the device pedal weight; *T_a[Nm]* for the torque generated due to motions dynamics; *Tao_B[Nm]* for the torque generated by the brake action; *Tao_H[Nm]* for the human torque seen on Equation (13); and *ang_accel[rad/*$s^{2}$*]* for the system acceleration.

For the controller dynamic observed in Equation (17), its implementation within the rehabilitation device is seen in Figure 13 as the *Fn_activate_controller* and *PI_Controller* blocks (red box with dotted lines), which receive the error signal and modify an activation function to regulate the voltage supply and generate a controlled $\tau_{B}$, as seen in Figure 14.

In Table 1, a 6-case scenario for different values for the controller parameters $K_{p}$ and $K_{i}$ applied to the control subsystem seen in Figure 14 is shown. Results of the different tests are presented in Appendix A.

The cases shown in Appendix A demonstrate that only in and scenario where $K_{p}$ is greater than $K_{i}$ does the response of the system satisfy the purpose of the speed controller. The following Figure 15 shows the dynamic system response when the scenario F is executed—the *x*-axis represents time in seconds and the *y*-axis represents the angular position in degrees—when the patient’s leg extension speed (red, sloped) exceeds the reference speed limit (black, horizontal) and how the control action forced the leg speed dynamics to reduce each time the threshold is exceeded.

The use of controllers to regulate the speed of human joint movements can generate certain ripples in the speed signal that affect the smoothness of the movements. However, these ripples can be smoothed out by using first- and second-order filters, which can be adjusted within the system bandwidth. In this way, sudden accelerations in the human joint can be prevented from interfering with the natural and fluid movement of the body, thus improving the user experience and the precision of controlled movement.

## 4. Conclusions

After carrying out the development of a technique for a control system that guarantees isokinetic control in rehabilitation devices for the recovery of muscle injuries in human legs, we reach the following conclusions.

Viewing the knee joint and the adjunct sections of the leg as simplified geometric figures, i.e., as triangles, facilitates the understanding of the ligament and muscle junction interactions used in the flexion and extension moments of the leg.

The speed control system used in isokinetic rehabilitation is the key to ensuring the safety and effectiveness of the treatment. This system uses a particle magnetic brake that accurately adjusts the rotation speed of the pulley. For purpose-built speed control systems, computational simulation tools that reflect the dynamic characteristics of the subsystems present in the medical device are necessary, which is reflected in a reduction of physical resources when dealing with testing methods to develop a digital dynamic system comparable to physical device prototypes. The constant rotational speed enabled by the speed control system is essential to accurately measure the torque generated in a determinate time by the human muscles.

The use of the proportional integrator controller for the speed control system enhances the response of the system to variations taken in early moments and during the time period of the exercise compared to the application of only a proportional-type control element, resulting in a quick response to both fast and slow changes in control action to alter the patient’s extension speed without compromising the tendon response, and providing organized and efficient information for further biomedical analysis.

Isokinetic rehabilitation is a technique that promises to be effective in the recovery of leg muscle injuries. The physical variables of elements involved in the mechanical system must be considered, such as the patient’s anthropometric variables, the physical variables of the mechanical and electrical components of the device, and the dynamics in which the bodies interact. These results are valuable for guiding the treatment of patients with leg muscle injuries and improving the effectiveness of isokinetic rehabilitation. The accurate strength data collected during the rehabilitation process will provide healthcare professionals with precise information. This will enable them to make more accurate diagnoses and make informed decisions about future procedures. By using reliable measurements of muscle strength and human torque, healthcare providers can customize treatment plans for each patient, optimizing their recovery and improving the overall outcomes of isokinetic rehabilitation. These findings contribute to the growing evidence supporting the importance of accurate data in patient care and treatment success.

Biomedical devices, such as the isokinetic rehabilitation system, are excellent measurement devices that help to collect data for patient evolution studies; in addition to the guidelines recommended by medical professionals, they allow for improvement to be observed when a maximum permissible speed is guaranteed during the extension process, which, in turn, can be observed by analyzing human torque and the force exerted by the anatomical elements responsible for extension. These results could be useful to guide the treatment of patients with muscular leg injuries and to improve the effectiveness of isokinetic rehabilitation in general.

## Figures and Tables

**Figure 1 sensors-23-05827-f001:**
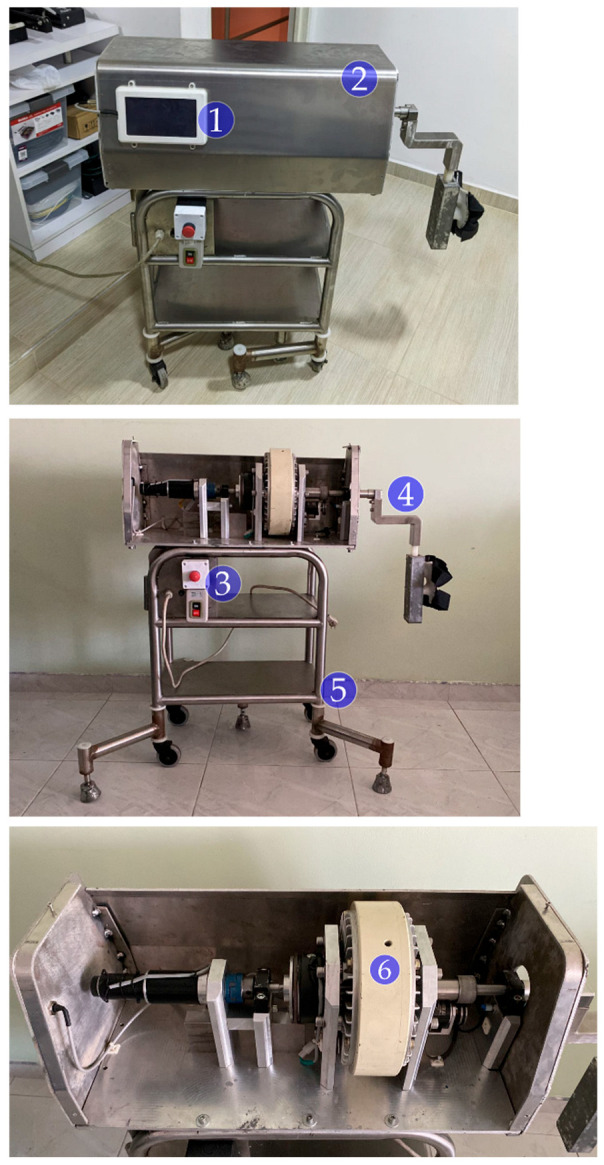
Rehabilitation device configured for a lower limb rehabilitation system from Universidad del Valle: (**1**) control screen; (**2**) chassis; (**3**) emergency; (**4**) foot pedal; (**5**) support structure stop; and (**6**) magnetic brake.

**Figure 2 sensors-23-05827-f002:**
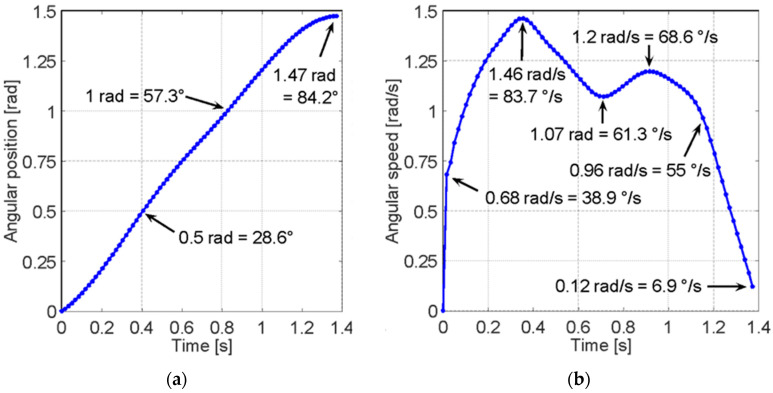
The behavior of the lower limb during extension. (**a**) Angular position at the knee joint for an uncontrolled extension. A tendency for the linear growth of extension can be observed from 0 radians to approximately 1.5 radians over time. (**b**) The angular speed at the knee joint during an uncontrolled leg extension. Multiple peaks can be observed during the extension of the leg reaching approximately 1.5 radians per second at the maximum speed segment.

**Figure 3 sensors-23-05827-f003:**
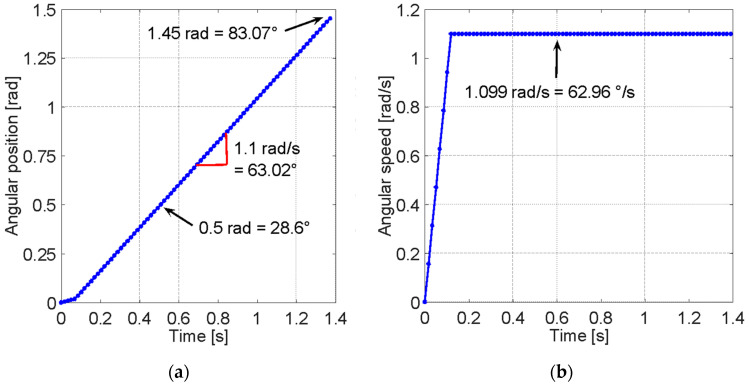
The behavior of the lower limb during extension. (**a**) Angular position at the knee joint for an ideal controlled extension. A better tendency for linear growth in the extension over time can be observed from 0 rad to approximately 1.5 rad. (**b**) The angular speed at the knee joint during an ideal controlled leg extension. A linear growth trend in extension speed is observed at the onset of motion, reaching stability at approximately 1.1 rads.

**Figure 4 sensors-23-05827-f004:**
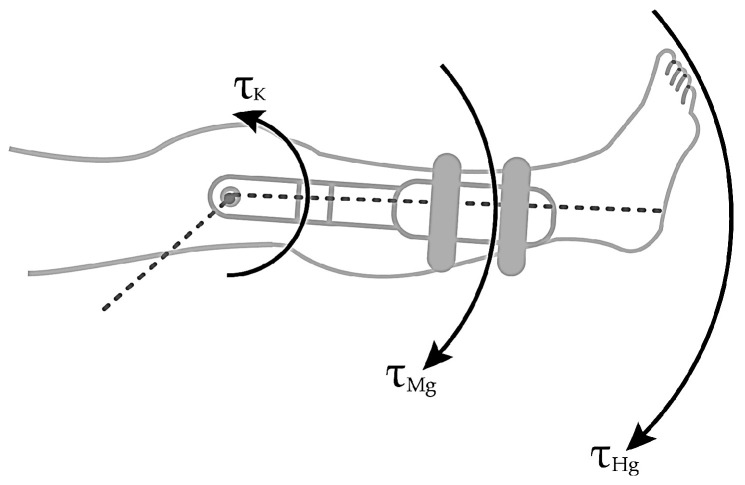
Torques are associated with knee joint extension.

**Figure 5 sensors-23-05827-f005:**
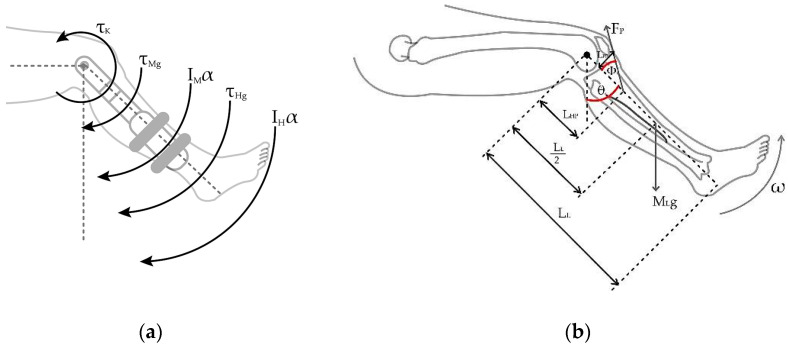
The dynamics of a lower limb during the extension. (**a**) Torques in the knee joint for maximum extension. (**b**) Diagram of forces and distances in the human leg.

**Figure 6 sensors-23-05827-f006:**
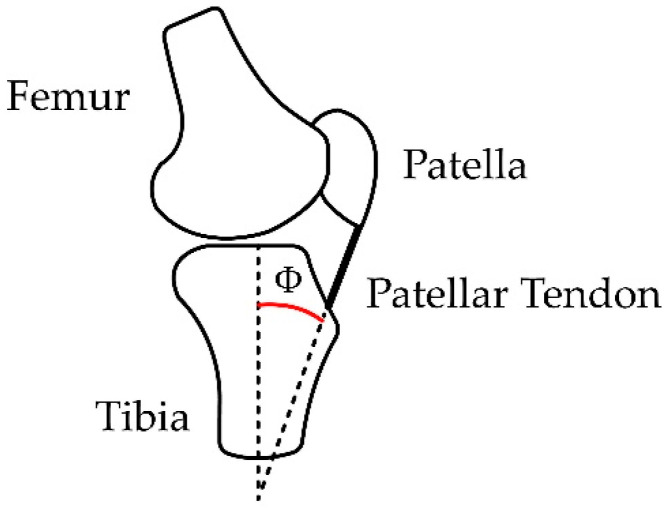
Presence of the patellar tendon and the separation angle between the tibia and the tendon.

**Figure 7 sensors-23-05827-f007:**
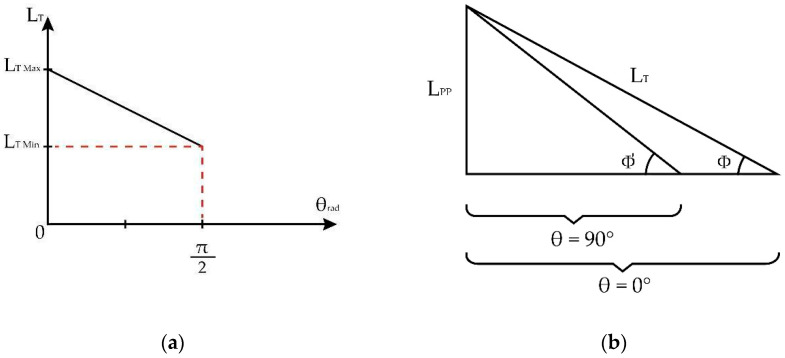
Measurements and distances for calculating Φ measured in radians due to the leg extension. (**a**) Relation between $L_{T}$ with respect to the leg extension angle. (**b**) Φ location regarding the tendon length during the leg extension. Variation of Φ according to the leg extension from 0 to 90°.

**Figure 8 sensors-23-05827-f008:**
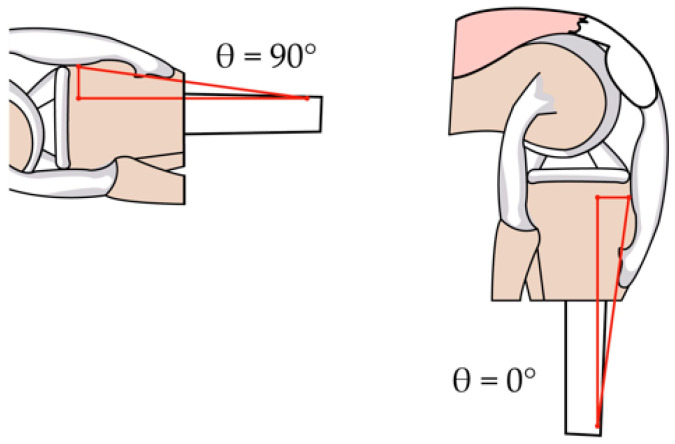
View of the triangle formed by the patellar tendon at the positions of maximum extension, θ=90° (1.57 radians), and reference flexion, θ=0° (0 radians).

**Figure 9 sensors-23-05827-f009:**
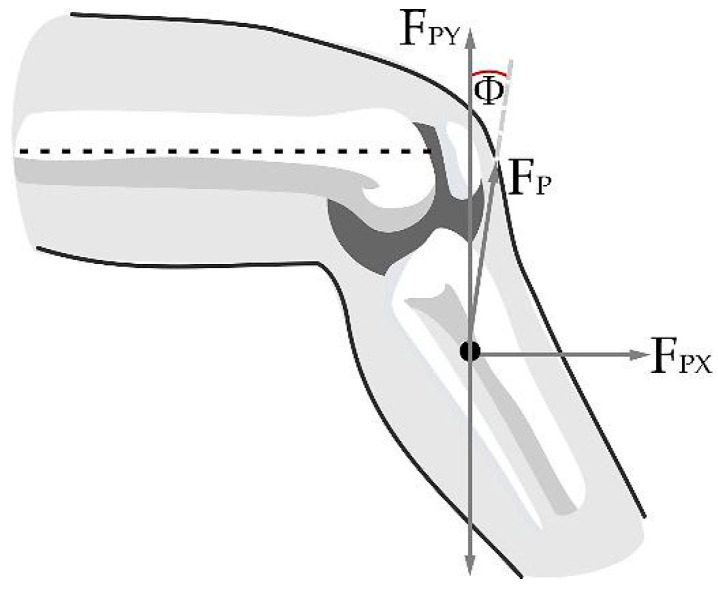
Cartesian components with respect to patellar tendon force.

**Figure 10 sensors-23-05827-f010:**
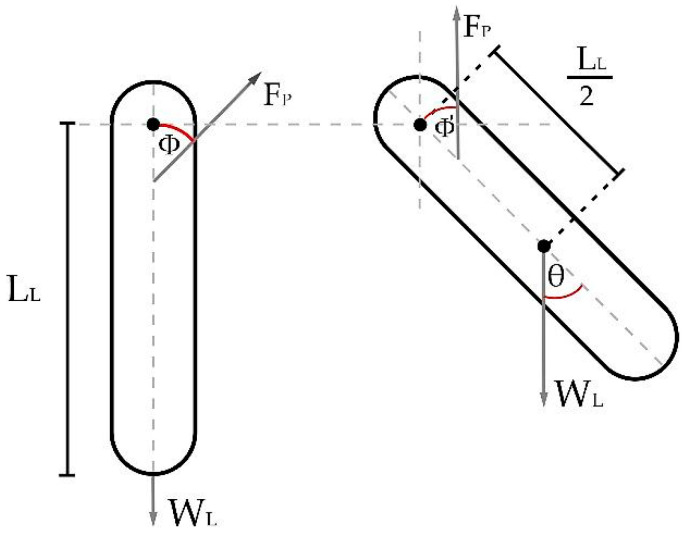
Reduction of components for human torque calculation.

**Figure 11 sensors-23-05827-f011:**
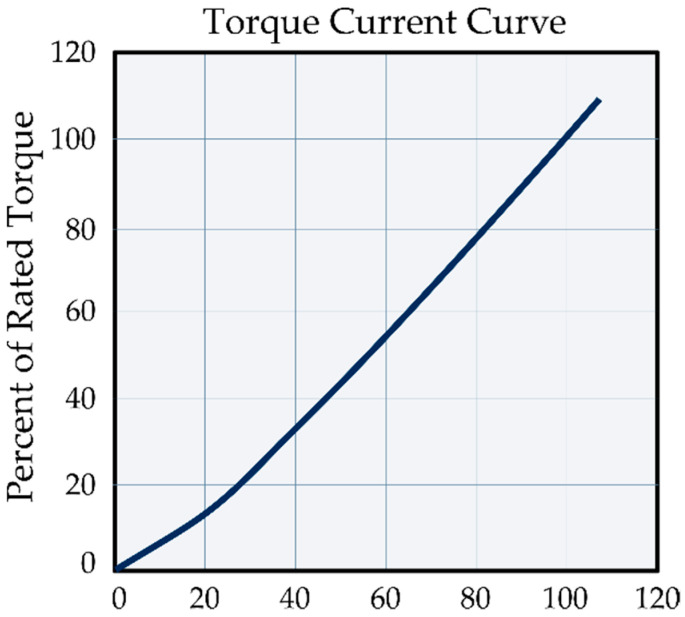
Average current percentage vs. average torque percentage for a GXFZ-B-6 model [34].

**Figure 12 sensors-23-05827-f012:**
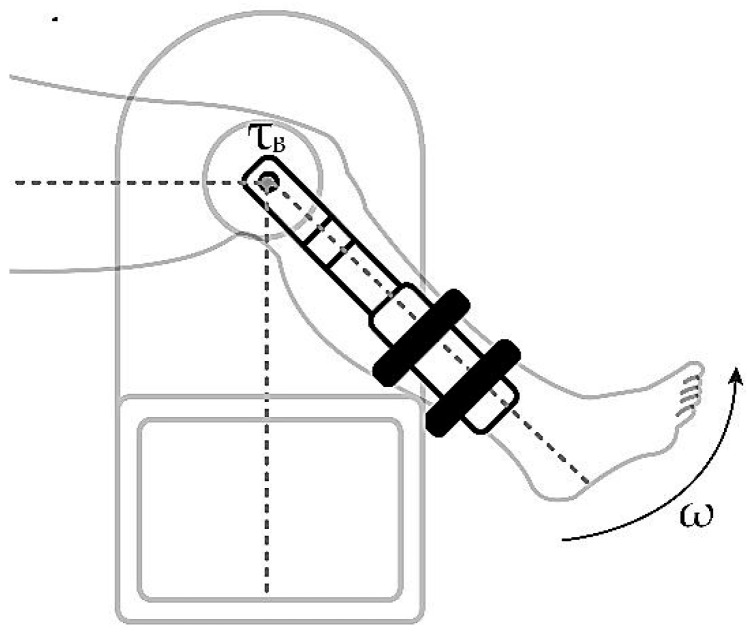
Patient’s leg attached to the rehabilitation device.

**Figure 13 sensors-23-05827-f013:**
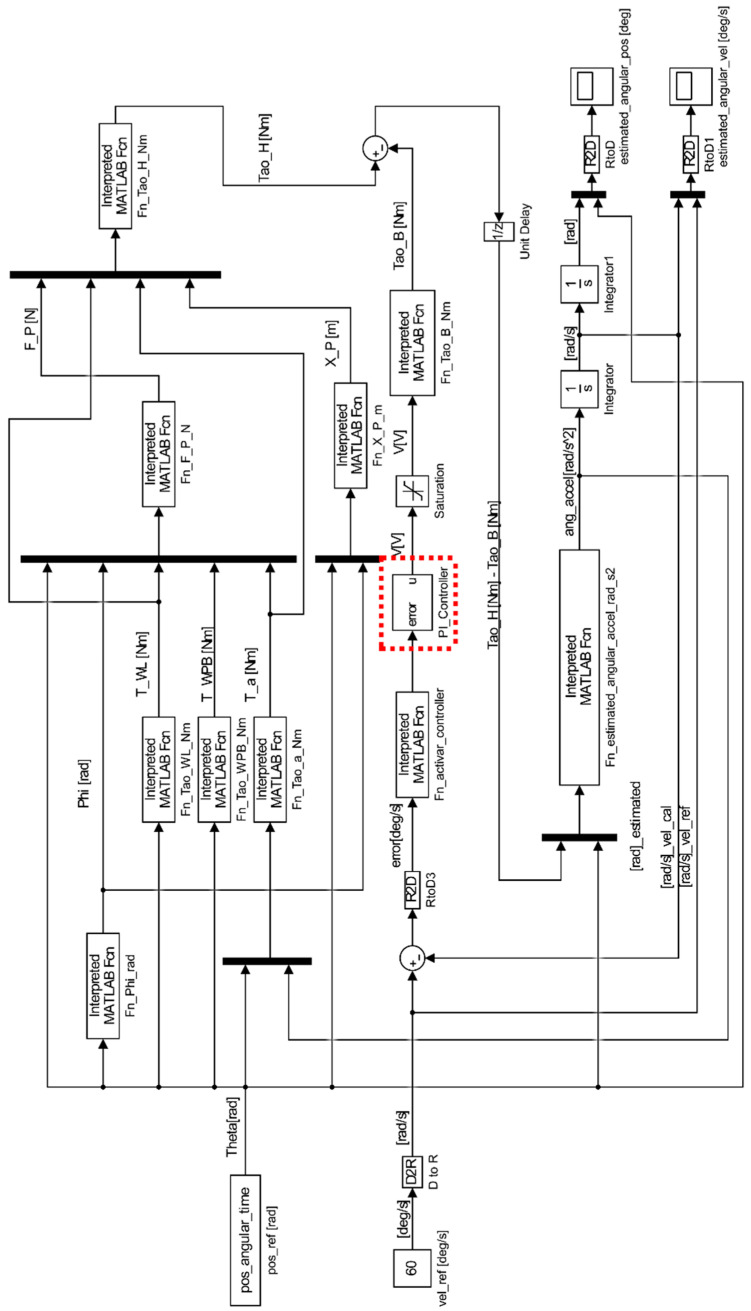
Dynamic modeling for the human–machine system in a closed loop.

**Figure 14 sensors-23-05827-f014:**
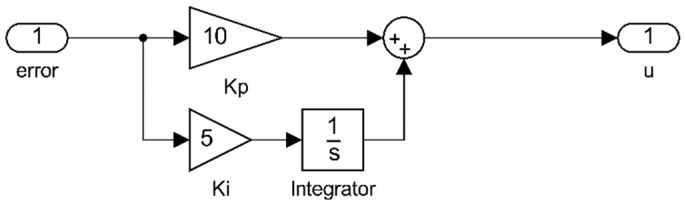
Control system implemented with proportional and integrative control.

**Figure 15 sensors-23-05827-f015:**
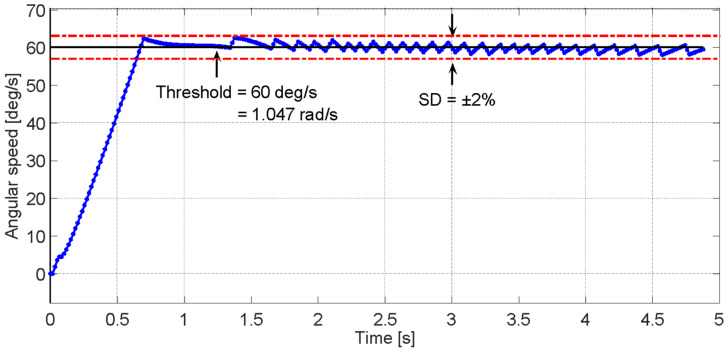
Time response of the dynamic system when controller A is applied.

**Table 1 sensors-23-05827-t001:** Different parameters for $K_{p}$ and $K_{i}$ implemented for the control system seen in Figure 14.

Case	$K_{p}$	$K_{i}$
A	0	0
B	5	0
C	0	5
D	5	5
E	5	10
F	10	5

## Data Availability

The datasets used and analyzed during the current study are available from the corresponding author on reasonable request.

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
