# Peer review of "Development of a Control Strategy in an Isokinetic Device for Physical Rehabilitation"

_sensors, 2023, doi:10.3390/s23135827_

Round 1

Reviewer 1 Report

General:

After reading the manuscript in its entirety, I come to the conclusion that much of this paper is above my head despite being familiar with the implementation of isokinetic dynamometry in sports medicine settings. I provide this review humbly, and hope that someone more well versed in physics and bioengineering has been recruited for peer-review.

The structure of this manuscript is atypical, (separate introduction and ‘literature review’ etc.). Perhaps this is typical in the field of bioengineering, but it is not for me. I would suggest the authors order the manuscript in a more traditional format.

While reading the middle sections, I am left wondering what the point of the article is. What are the authors trying to convey? That they can/have developed a better dynamometer system? Better how? What are they proposing that cannot be already done? I am sorry if I am just not well versed, or perhaps smart enough to understand, but I honestly am lost here.

Title:

I am not sure what is meant by ‘control strategy’

Abstract:

I found the abstract a bit hard to follow. From the abstract in isolation, I am not able to tell what the device is doing, or what is meant by ‘control strategy’ in the title.

It feels like the authors are using several ‘fancy’ words to make the paper sound more interesting. However, I believe this has the opposite effect as I am left confused. I highly recommend that the authors attempt to use more layman language to make the abstract more understandable to most readers.

Introduction:

The introduction is 3 paragraphs, but has only 1 citation. There are plenty of examples of sentences that should include references. For example, the authors write:

“It has been shown to be effective in the recovery of leg muscles injuries, allowing patients to exercise safely with controlled intensity, which facilitates the recovery of muscle strength and endurance. This technique can also help prevent muscle wasting and atrophy in patients with physical disabilities or prolonged periods of inactivity.”

But provide no references to works demonstrating these findings…

The introduction never really introduces a problem that that research is aiming to address. Isokinetic contractions/devices are briefly discussed, but I have little indication of the issue with traditional isokinetic dynamometers, and why the new device is needed/warranted. This is a major issue that the authors must be sure to address here, and throughout the manuscript (including the abstract).

Literature review:

While the authors make fine points, I find that like the introduction, there is a lack of supporting studies cited.

The second paragraph of the section also states that isokinetic assessment is better than isometric because it can be used to measure ‘agonist-antagonist relationships, determine maximum strength, evaluate endurance and other indicators of muscle capacity. However, isometric dynamometry can measure all these variables as well, and in the case of maximum strength, perhaps even better (and more reliably) than isokinetic contractions…

What is meant by “are insufficient to apply a complete isokinetic rehabilitation protocol, as they do not include all the necessary components for a complete treatment.” Can the authors please elaborate? Are these devices missing hardware, software, user knowledge etc.?

Paragraph 6 of this section starts with “The study…” this is very confusing and not how a paragraph should be started. In general, the writing could be improved substantially.

Results and Discussion:

I am once again wondering what the point of this all is. It would have been fascinating if the authors had compared contractions in a commercial isokinetic device with contractions in this system, but as far as I can tell, they have not.

Therefore, I cannot say what the point of this endeavor was.

The authors also write:

“These results could be useful to guide the treatment of patients with muscular leg injuries and to improve the effectiveness of isokinetic rehabilitation in general.”

However, I do not get a sense of how/why this might be the case.

The English is not bad but could be improved in some spots. I believe my confusion is more because of content, not writing skills.

Author Response

  • Responses to reviewer No. 1

Comments and Suggestions for Authors: 

General:

After reading the manuscript in its entirety, I come to the conclusion that much of this paper is above my head despite being familiar with the implementation of isokinetic dynamometry in sports medicine settings. I provide this review humbly, and hope that someone more well versed in physics and bioengineering has been recruited for peer-review.

R: A general style editing was carried out. Additionally, references in the field of isokinetic rehabilitation were incorporated, such as [1] and [3], to support the relevance of the use of isokinetic rehabilitation devices in various applications, such as sports medicine and post-surgical treatments.

The structure of this manuscript is atypical, (separate introduction and ‘literature review’ etc.). Perhaps this is typical in the field of bioengineering, but it is not for me. I would suggest the authors order the manuscript in a more traditional format.

R: A paragraph describing the document’s structure is added at the end of the introduction. A literature review was also integrated into the introduction section to provide a traditional format.

While reading the middle sections, I am left wondering what the point of the article is. What are the authors trying to convey? That they can/have developed a better dynamometer system? Better how? What are they proposing that cannot be already done? I am sorry if I am just not well versed, or perhaps smart enough to understand, but I honestly am lost here.

R: According to the literature review, this work aims to define a control strategy by implementing a commonly used approach in industrial applications, namely a proportional-integral controller. A dynamic model of the coupled human-rehabilitation device system was developed to tune the controller parameters. Specifically, the authors present a paper with the purpose of establishing an approach that allows for the definition, tuning, and evaluation of a control strategy through the interaction of the coupled system.

Title:

I am not sure what is meant by ‘control strategy’

R: Considering that a control strategy in a dynamic system aims to generate a control effort to properly regulate a variable of interest towards a desired reference, which is the angular velocity of the user's knee joint. The control objective proposed in work is to vary the mechanical impedance to achieve a constant rotational speed, allowing for the implementation of a feasible isokinetic protocol for knee rehabilitation.

Abstract:

I found the abstract a bit hard to follow. From the abstract in isolation, I am not able to tell what the device is doing, or what is meant by ‘control strategy’ in the title. It feels like the authors are using several ‘fancy’ words to make the paper sound more interesting. However, I believe this has the opposite effect as I am left confused. I highly recommend that the authors attempt to use more layman language to make the abstract more understandable to most readers.

R: In addition to the previous response, a general style editing of the abstract was carried out, strengthening the contextualization regarding the application of isokinetic rehabilitation protocols using technical language appropriate to the integrated research area of biomechanics, control systems, and instrumentation. Additionally, a paragraph was added to clarify the article's contribution. Finally, it is clarified that the consolidation of this control strategy within the context of the work offers the opportunity for the integration of such physical rehabilitation therapies with a teleoperated infrastructure, thereby enhancing the provision of physical rehabilitation telemedicine programs.

Introduction:

The introduction is 3 paragraphs, but has only 1 citation. There are plenty of examples of sentences that should include references. For example, the authors write:

“It has been shown to be effective in the recovery of leg muscles injuries, allowing patients to exercise safely with controlled intensity, which facilitates the recovery of muscle strength and endurance. This technique can also help prevent muscle wasting and atrophy in patients with physical disabilities or prolonged periods of inactivity.”

But provide no references to works demonstrating these findings…

The introduction never really introduces a problem that that research is aiming to address. Isokinetic contractions/devices are briefly discussed, but I have little indication of the issue with traditional isokinetic dynamometers, and why the new device is needed/warranted. This is a major issue that the authors must be sure to address here, and throughout the manuscript (including the abstract).

R: A general style of editing was carried out to provide greater clarity regarding the context of the work. In this regard, several relevant statements expressed in the introduction were referenced.

Literature review:

While the authors make fine points, I find that like the introduction, there is a lack of supporting studies cited.

The second paragraph of the section also states that isokinetic assessment is better than isometric because it can be used to measure ‘agonist-antagonist relationships, determine maximum strength, evaluate endurance and other indicators of muscle capacity. However, isometric dynamometry can measure all these variables as well, and in the case of maximum strength, perhaps even better (and more reliably) than isokinetic contractions…

What is meant by “are insufficient to apply a complete isokinetic rehabilitation protocol, as they do not include all the necessary components for a complete treatment.” Can the authors please elaborate? Are these devices missing hardware, software, user knowledge etc.?

Paragraph 6 of this section starts with “The study…” this is very confusing and not how a paragraph should be started. In general, the writing could be improved substantially.

R: Regarding the first paragraph, additional statements from studies supporting the applications of isokinetic rehabilitation devices were included.

Regarding the second paragraph, the text was edited to incorporate the suggestions made by the evaluator.

Regarding the third and fourth paragraphs, a general style editing was conducted.

Results and Discussion:

I am once again wondering what the point of this all is. It would have been fascinating if the authors had compared contractions in a commercial isokinetic device with contractions in this system, but as far as I can tell, they have not.

Therefore, I cannot say what the point of this endeavor was.

R: A reference was made to commercial products. In paragraph 7, a literature review was conducted to demonstrate the advantages of applying isokinetic exercises. Additionally, in paragraph 12, it is explained that current commercial solutions for isokinetic dynamometry lack reliability in measuring flexor torques for the knee.

The authors also write:

“These results could be useful to guide the treatment of patients with muscular leg injuries and to improve the effectiveness of isokinetic rehabilitation in general.”

However, I do not get a sense of how/why this might be the case.

R: In the fifth paragraph of the conclusions, a description is added about using the collected variables during the rehabilitation process with the proposed device and how they provide better data for future analysis by healthcare personnel.

Comments on the Quality of English Language: The English is not bad but could be improved in some spots. I believe my confusion is more because of content, not writing skills.

R: The quality of the English language was enhanced, resulting in improved clarity and effectiveness of communication.

Reviewer 2 Report

This is a very interesting work, and the research content is rich. The approach is interesting, and the presentation of the theoretical aspects are very good.

The topic of the article is important and interesting. It was well prepared. There is a tendency of the authors to consider issues relevant to the topic being pursued.

The graphical presentation of the results, in the form of tables and graphics, enriched the presentation.

The only observation is related to the mechanical structure of the device showed in figure 1: it is insufficiently described or represented

Author Response

  • Responses to reviewer No. 2

Comments and Suggestions for Authors: This is a very interesting work, and the research content is rich. The approach is interesting, and the presentation of the theoretical aspects are very good.

The topic of the article is important and interesting. It was well prepared. There is a tendency of the authors to consider issues relevant to the topic being pursued.

The graphical presentation of the results, in the form of tables and graphics, enriched the presentation.

The only observation is related to the mechanical structure of the device showed in figure 1: it is insufficiently described or represented.

R: Paragraphs 4 and 5 in the introduction have been included to provide concise information regarding the structural design of the device. Furthermore, an additional image has been incorporated to represent the machine in an open configuration visually.

Reviewer 3 Report

To achieve physical recovery from musculoskeletal injuries in the lower limbs and achieve the reintegration of the affected subject into society as an independent and autonomous individual in their daily activities, this paper presents a control model that introduces a medical isokinetic rehabilitation protocol. However, the following issues need to be addressed before this paper can be published.

1.    In the Abstract, before introducing the Isokinetic rehabilitation, the author should use several sentences to introduction the background of the rehabilitation. Most references are too old, and please cite some recent works such as, such as “Development of an Untethered Adaptive Thumb Exoskeleton for Delicate Rehabilitation Assistance, IEEE Transactions on Robotics, 2022, 38(6): 3514-3529” and “A Miniature Elastic Torque Sensor for Index Finger Exoskeletons, IEEE Transactions on Instrumentation and Measurement, 2023”, etc.

2.     The texts in the Figures are too small. It is better to show them in a unified way.

3.     Some formulas such as “Nm/A” should be shown in a standardized way.

4.     The expression of “Where,” should be revised to “where,”

 Minor editing of English language required

Author Response

  • Responses to reviewer No. 3

Comments and Suggestions for Authors: To achieve physical recovery from musculoskeletal injuries in the lower limbs and achieve the reintegration of the affected subject into society as an independent and autonomous individual in their daily activities, this paper presents a control model that introduces a medical isokinetic rehabilitation protocol. However, the following issues need to be addressed before this paper can be published.

  1. In the Abstract, before introducing the Isokinetic rehabilitation, the author should use several sentences to introduction the background of the rehabilitation. Most references are too old, and please cite some recent works such as, such as “Development of an Untethered Adaptive Thumb Exoskeleton for Delicate Rehabilitation Assistance, IEEE Transactions on Robotics, 2022, 38(6): 3514-3529” and “A Miniature Elastic Torque Sensor for Index Finger Exoskeletons, IEEE Transactions on Instrumentation and Measurement, 2023”, etc.

R: First, the context of the assisted rehabilitation of the lower limbs through robotic systems, and the main aim of the rehabilitation, were added at the beginning of the abstract. Second, a stylistic edit was made to the third paragraph of the abstract to clarify the matter.

  1. The texts in the Figures are too small. It is better to show them in a unified way.

R: The figures that present signals over time were improved in resolution and the size of the

embedded texts. The information presented in the images was enhanced by updating the device views. Additionally, the graphs’ font type was consistent with the font type used in the written content, and a background was added to the number of balloons to improve their visibility.

  1. Some formulas such as “Nm/A” should be shown in a standardized way.

R: The motor constant of the magnetic dipole, K_B, was defined using the engineering expression that relates the torque rate to the units of Ampere.

  1. The expression of “Where,” should be revised to “where,”

R: The expressions "Where" or "where" have been replaced with suitable connectors.

Comments on the Quality of English Language: Minor editing of English language required

R: English-style editing was applied at specific points to improve reading comprehension.

Round 2

Reviewer 1 Report

I commend the authors for taking my critique to heart and working to improve the manuscript. They have done an excellent job regarding all of my comments. While the practical application of the control system is still questionable in my mind, the fact that the present journal is not aimed at endpoint users makes me fine with endorsing publication at this time.

Well done.